# Multi-omic analysis of stroke recurrence in African Americans from the Vitamin Intervention for Stroke Prevention (VISP) clinical trial

**Nicole M. Davis Armstrong**[1☉], **Kelsey J. Spragley**[1☉], **Wei-Min Chen**[2,3], **Fang-Chi Hsu**[4], **Michael S. Brewer**[1], **Patrick J. Horn**[1], **Stephen R. Williams**[5], **Michèle M. Sale**[2,3], **Bradford B. Worrall**[3,5], **Keith L. Keene**[1,6]*

1 Department of Biology, East Carolina University, Greenville, North Carolina, United States of America,
2 Center for Public Health Genomics, University of Virginia, Charlottesville, Virginia, United States of America,
3 Department of Public Health Sciences, University of Virginia, Charlottesville, Virginia, United States of America, 4 Division of Public Health Sciences, Department of Biostatistics and Data Science, Wake Forest School of Medicine, Winston-Salem, North Carolina, United States of America, 5 Department of Neurology, University of Virginia, Charlottesville, Virginia, United States of America, 6 Center for Health Disparities, Brody School of Medicine, East Carolina University, Greenville, North Carolina, United States of America

☉ These authors contributed equally to this work.
* keenek@ecu.edu

## Abstract

African Americans endure a nearly two-fold greater risk of suffering a stroke and are 2–3 times more likely to die from stroke compared to those of European ancestry. African Americans also have a greater risk of recurrent stroke and vascular events, which are deadlier and more disabling than incident stroke. Stroke is a multifactorial disease with both heritable and environmental risk factors. We conducted an integrative, multi-omic study on 922 plasma metabolites, 473,864 DNA methylation loci, and 556 variants from 50 African American participants of the Vitamin Intervention for Stroke Prevention clinical trial to help elucidate biomarkers contributing to recurrent stroke rates in this high risk population. Sixteen metabolites, including cotinine, N-delta-acetylornithine, and sphingomyelin (d17:1/24:1) were identified in t-tests of recurrent stroke outcome or baseline smoking status. Serum tricosanoyl sphingomyelin (d18:1/23:0) levels were significantly associated with recurrent stroke after adjusting for covariates in Cox Proportional Hazards models. Weighted Gene Co-expression Network Analysis identified moderate correlations between sphingolipid markers and clinical traits including days to recurrent stroke. Integrative analyses between genetic variants in sphingolipid pathway genes identified 29 nominal associations with metabolite levels in a one-way analysis of variance, while epigenomic analyses identified xenobiotics, predominately smoking-associated metabolites and pharmaceutical drugs, associated with methylation profiles. Taken together, our results suggest that metabolites, specifically those associated with sphingolipid metabolism, are potential plasma biomarkers for stroke recurrence in African Americans. Furthermore, genetic variation and DNA methylation may play a role in the regulation of these metabolites.

**Data Availability Statement:** Individual level VISP genetic, metabolomics, and epigenetics data is considered sensitive controlled data and cannot be shared publicly, as specified by the IRBs of Wake Forest University School of Medicine, the University of North Carolina at Chapel Hill School of Medicine, and the University of Virginia School of Medicine, along with the NIH Data Access Committee. Controlled-access data can only be obtained if a user has been authorized by the appropriate Data Access Committee (DAC). The individual level Genomics and Randomized Trials Network (GARNET) VISP data are available in the database of Genotypes and Phenotypes (dbGaP) (Accession: phs000343.v3.p1) and can be requested through the dbGaP Authorized Access System (https://dbgap.ncbi.nlm.nih.gov/aa/wga. cgi?page=login), the Cerebrovascular Disease Knowledge Portal (https://cd.hugeamp.org/), and the Coriell Institute for Medical Research (https:// catalog.coriell.org/). The authors will also share the data on request.

**Funding:** This work was supported by the National Institutes of Health, grant U01HG005160 and Supplement U01HG005160-03S1 (MMS/BBW) from the National Human Genome Research Institute, grant R15 NS095115-01A1 and Supplement R15NS095115-01A1S1 (KLK/MSB) from the National Institute of Neurological Disorders and Stroke, American Heart Association Scientist Development Award 12SDG9180012 (KLK), and funds from the Walking for My Life 5K Stroke Walk sponsored by Faith Christian Center International (Charlottesville, VA). The original study recruitment and datasets for the VISP clinical trial were supported by research grant R01 NS34447 (JT) from the National Institute of Neurological Disorders and Stroke.

**Competing interests:** The authors have declared that no competing interests exist.

## Introduction

Stroke is a leading cause of death and disability globally, resulting in more than 6 million deaths per year [1]. For reasons yet to be elucidated, individuals of African descent are more likely to suffer a stroke, have an earlier age of occurrence, and experience poorer outcomes compared to individuals of other ancestry [2]. While overall stroke mortality rates have declined since 1990, the rates have remained higher among African Americans (AAs) than any other racial group in the United States. For example, stroke mortality rates are approximately 50% higher in AAs than their white counterparts with this mortality increasing with age (e.g. at age 55 the risk is up to three times greater) [3]. While racial disparities are prevalent in primary strokes, data on AAs and stroke recurrence are rare, even though recurrent strokes are more disabling and deadlier than initial strokes and account for 25% of annual cases [1]. The management of stroke risk factors, including hypertension and diabetes, remains an important strategy to decrease the stroke burden.

The use of blood biomarkers that aid in stroke diagnosis and early identification of patients with high-risk of recurrence has been difficult to establish [4]. Part of this analytical complexity derives from the biologically complex set of associated effects, which can vary from person to person depending on the type of stroke, its severity, the location within the brain, number of strokes and recurrence, and other general health parameters. One of the most common approaches is to utilize a comparative analysis of core metabolites (e.g. amino acids, lipids, carbohydrates, coenzymes) quantified by mass spectrometric-based profiling of serum samples from stroke patients versus appropriate controls. Indeed, recent studies have identified several aberrant serum and urine metabolic biomarkers associated with stroke. For example, in a metabolomics analysis of 3,900 participants from the Atherosclerosis Risk in Communities Study (ARIC), two serum long-chain dicarboxylic acids, tetradecanedioate and hexadecanedioate (hazard ratio [95% confidence interval (CI)] 1.11 [1.06–1.16] and 1.12 [1.07–1.17], respectively), were associated with incident ischemic stroke [5]. In contrast, a smaller study of 40 patients with acute ischemic stroke more broadly identified potential metabolic biomarkers spanning amino acid, fatty acid, phospholipid, and choline metabolism pathways [6]. In a separate study of 67 cerebral infarction patients, metabolites that comprise the folate one-carbon cycle, including folic acid, cysteine, S-adenosyl homocysteine, and oxidized gluthaione, were determined as potential stroke biomarkers using an ultra-high-performance liquid chromatography in tandem with time of flight mass spectrometer (UPLC–TOF MS)-based metabolomic approach [7]. These few studies collectively demonstrate the complexity of analyzing metabolic events associated with stroke and identification of reliable biomarkers.

In this current study, we conducted an integrative -omics analysis utilizing serum metabolites, DNA methylation, and genotype data from 50 African American participants from the Vitamin Intervention for Stroke Prevention (VISP) clinical trial. Like the aforementioned studies, metabolomics analyses of serum from stroke patients revealed additional metabolic biomarkers associated with recurrent stroke events. However, subsequent integration with genetics data has provided additional confidence in these event-metabolome associations and provides a foundation for future studies.

## Methods

### Ethics statement

The VISP clinical trial study protocol was approved by the institutional review boards (IRBs) of Wake Forest University School of Medicine, the University of North Carolina at Chapel Hill School of Medicine, and individual recruitment sites. All VISP participants provided written,

informed consent and a subset of 2,100 participants agreed to be included in subsequent genetic studies. IRB approval from the University of Virginia and East Carolina University was obtained for the genetic, epigenetic, and metabolomic components.

## The Vitamin Intervention for Stroke Prevention (VISP) trial

The Vitamin Intervention for Stroke Prevention (VISP) clinical trial was a multi-centered, double-blinded, randomized and controlled clinical trial designed to determine whether a combination pyridoxine (vitamin B6), cyanocobalamin (vitamin B12), and folic acid (vitamin B9) supplementation reduced recurrent cerebral infarction, myocardial infarction (MI), or fatal coronary heart disease [8]. Participants were enrolled within 120 days of suffering a non-disabling cerebral infarction, assigned a daily high-dose or low-dose B-vitamin formulation, and followed for two years. VISP inclusion/exclusion criteria are listed below.

Inclusion criteria included:

1. Participants aged 35 years or older with elevated baseline homocysteine levels at or above the 25th percentile;

2. Participant enrollment within 120 days of suffering a non-disabling cerebral infarction characterized by the sudden onset of a neurological deficit persisting at least 24 hours and observed on CT or MRI;

3. Geographical accessibility for follow-up;

4. Adequate means of transportation;

5. Compliance of 75% or greater with vitamin regiment in the run-in period and

6. Patient agreement to take study medication and avoid other vitamin supplements containing folic acid and B6

VISP exclusion criteria included:

1. Stroke due to any form of intracranial hemorrhage, dissection of a cervico-cephalic artery, veno-occlusive disease, drug abuse, or vasculitis;

2. CT or MRI of brain showing lesion other than ischemic infarction as cause;

3. Modified Rankin Stroke Scale (RSS) score of 4–5 at time of eligibility determination;

4. Presence of specific potential sources of cardiogenic emboli, such as atrial fibrillation within 30 days of stroke or history of prosthetic cardiac valve, intracardiac thrombus or neoplasm, or valvular vegetation;

5. Presence of major neurologic illness apart from stroke that would prevent proper evaluation of recurrent stroke;

6. Presence of cancer, pulmonary disease, or other illness which, in the opinion of the study physician, would limit the life expectancy of the patient to less than two years;

7. Severe congestive heart failure;

8. Renal insufficiency requiring dialysis;

9. Untreated pernicious anemia or untreated B12 deficiency;

10. Uncontrolled hypertension defined as systolic blood pressure >185 mm or diastolic >105 mm on two readings separated by five minutes at time of eligibility determination;

11. Conditions that prevent reliable participation in the study, such as refractory depression, severe cognitive impairment, alcoholism, or other substance abuse;

12. Use of medications, within the last 30 days, that affect homocysteine, such as methotrexate, tamoxifen, L-dopa, or phenytoin or bile acid sequestrants that can decrease folate levels

13. Woman of childbearing potential, defined as not having reached natural or surgical menopause or having had tubal ligation;

14. Participation in another trial in which active intervention is being received;

15. Participants on multivitamin supplements or single vitamins of B6 or folic acid were excluded unless they were willing to take the study supplements in place of the one(s) they usually took and/or

16. Any surgical procedure requiring a general anesthesia or hospital stay of three days or more, any type of invasive cardiac instrumentation, or an endarterectomy, stent placement, thrombectomy, or any other endovascular treatment of an abnormal carotid artery performed within 30 days prior to randomization or scheduled to be performed within 30 days after randomization [8].

In this study, global metabolite data were generated from serum samples of 50 AAs, enriched for recurrent stroke cases (N = 28). When possible, nonrecurrent controls were matched with recurrent stroke cases for age (within eight years), sex, number of cigarettes smoked per day (within 5 cigarettes), and disability measured by the modified Rankin Stroke Scale (RSS; within two points) (S1 and S2 Tables).

## Stroke recurrence, vascular outcomes, and clinical trait definition

In this study, recurrent stroke was defined as an acute neurological ischemic event of at least 24 hours duration with focal signs and symptoms and without evidence of primary intracranial hemorrhage or alternate explanation. Additionally, one of the following was present: 1) a one-point increase in the NIH stroke scale (NIHSS) in a previously normal section or 2) a new or 3) extended abnormality seen on CT or MRI. Diagnoses were reviewed by a local neurologist, two endpoint reviewers, and on a case-by-case basis, a Stroke Endpoint Review Committee. Underlying cause of death was decided by a Death Review Committee composed of physicians independent of VISP. Decisions were based on information available from hospital records, death certificates, coroners' reports, or physicians' questionnaires. Ischemic stroke subtype information is not available [8, 9]. Composite vascular event was defined as fatal coronary heart disease, a nonfatal hospitalized myocardial infarction (MI), and resuscitation for cardiac collapse, coronary bypass surgery, coronary angioplasty, or VISP recurrent stroke. Fatal/disabling stroke, MI, or death (FDMD) was defined as an outcome variable which identified individuals who suffered a disabling recurrent stroke or MI during the clinical trial or those who died from any cause. All of these outcomes were expressed as dichotomous variables and days from trial randomization to vascular event were recorded as time to event.

The "recurrent stroke ever" variable indicated whether an individual had suffered a stroke in addition to the VISP enrollment stroke. The MI variable was indicative of an individual suffering a MI during the VISP clinical trial. Known stroke risk factors such as the number of cigarettes smoked per day, smoking status (at enrollment and ever), blood pressure (BP) measurements (systolic and diastolic), self-reported hypertension (HTN) status, diabetes status, the number of strokes prior to VISP, the treatment arm, age, sex, and body mass index

(BMI) were included in analyses. Additionally, four stroke severity/functional scales were included: Mini-Mental State (MMS) status scale, NIHSS, RSS, and Barthel Stroke Questionnaire Form (BAR).

## Metabolomics

Global, untargeted metabolic profiling of 922 metabolites for the 50 AA VISP participants was performed by Metabolon, Inc (Durham, NC) using gas chromatography-mass spectrometry and liquid chromatographic-mass spectrometry protocols, as previously described [10]. Metabolite levels were measured using the fasting serum plasma samples from the baseline VISP visit.

Supernatants from a methanol extraction were used for analyses by two reverse phase (RP) ultra-performance liquid chromatography tandem mass spectrometry (UPLC-MS/MS) methods with positive ion mode electrospray ionization (ESI); (RP) UPLC-MS/MS with negative ion mode ESI; and hydrophilic interaction liquid chromatography/UPLC-MS/MS with negative ion mode ESI. All methods utilized a Waters ACQUITY UPLC system (Waters Corporation, Milford, MA) and a Thermo Scientific Q-Exactive high resolution/accurate MS interfaced with a heated ESI source and Orbitrap mass analyzer (ThermoFisher Scientific, Waltham, MA). Raw data was extracted, peak-identified and processed using Metabolon's proprietary hardware and software. Compounds were identified by comparison to library entries of purified standards or recurrent unknown entities that contain the retention time/index, mass to charge ratio, and chromatographic data, including tandem MS spectral data, on all molecules present in the library (S1 Appendix). Metabolite peaks were quantified using area-under-the-curve, and a data normalization step was performed to correct variation resulting from instrument inter-day tuning differences, setting the medians equal to one. A log transformation was performed on the normalized data for subsequent analysis.

Instrument variability was determined by calculating the median relative standard deviation (RSD) for the internal standards that were added to each sample prior to injection into the mass spectrometers (median RSD = 3%). Overall process variability was determined by calculating the median RSD for all endogenous metabolites (median RSD = 9%). These values for instrument and process variability met Metabolon's acceptance criteria.

## DNA methylation

As previously described, Illumina Infinium HumanMethylation450K BeadChip microarrays (Illumina, Inc., San Diego, CA, United States) were used per the manufacturer's protocol to determine the degree of methylation (β values) for 485,512 methylation loci [11]. Quality control and quantile normalization was performed, resulting in 473,864 autosomal methylation CpG loci [12–14].

## Genetic association of selected variants

Previously, 2100 VISP participants were genotyped on the Illumina HumanOmni1-Quad_v1-0_B BeadChip (Illumina, Inc) [9, 15]. We selected 556 variants spanning ±10 kb of 24 genes associated with sphingolipid metabolism and the sphingomyelinase pathway (S3 Table). Individual level VISP genetic, metabolomics, and epigenetics data is considered sensitive controlled data and cannot be shared publicly, as specified by the IRBs of Wake Forest University School of Medicine, the University of North Carolina at Chapel Hill School of Medicine, and the University of Virginia School of Medicine, along with the NIH Data Access Committee. Controlled-access data can only be obtained if a user has been authorized by the appropriate Data Access Committee (DAC). The individual level Genomics and Randomized Trials

Network (GARNET) VISP data are available in the database of Genotypes and Phenotypes (dbGaP) (Accession: phs000343.v3.p1) and can be requested through the dbGaP Authorized Access System (https://dbgap.ncbi.nlm.nih.gov/aa/wga.cgi?page=login), the Cerebrovascular Disease Knowledge Portal (https://cd.hugeamp.org/), and the Coriell Institute for Medical Research (https://catalog.coriell.org/). The authors will also share the data on request.

## Statistical analysis

Baseline characteristics of the study participants with and without recurrent stroke were compared using t-tests and chi-squared ($\chi^2$) tests for continuous and categorical traits, respectively. Univariate Welch's two sample t-tests were performed using all VISP participants (N = 50) and compared recurrent stroke, recurrent stroke ever, composite vascular event, and smoking statuses. VISP recurrent stroke analyses were stratified by sex, treatment arm, diabetes status (DM), and/or smoking status (S4 Table). A Bonferroni correction accounting for the number of metabolites tested was applied to determine the significance threshold of p≤ 5.42e-05 (error rate of 0.05 divided by the total number of metabolites, or = 0.05/922). A suggestive threshold of p≤4.42e-04 (= 0.05/113 total number of metabolite sub-pathways) was also implemented. Statistical power was calculated as 0.786, assuming a large effect size (Cohen's d = 0.8) using the "pwr" package in R [16, 17].

Matched pair analyses on a subset of 44 VISP participants (22 recurrent, 22 non-recurrent matched on age, sex, cigarettes smoked per day, and enrollment stroke severity) was conducted utilizing a matched-pair t-test. Statistical and suggestive significance was calculated as p≤5.42e-05 (Bonferroni adjustment described above) and p≤ 2.27e-03 (= 0.05/22 matched pairs), respectively. Statistical power of 0.8 was reached using a two-sided test, an effect size of 0.626, alpha = 0.05, and n = 22 matched pairs [16, 17].

Cox proportional hazards (PHs) regression models were used to identify metabolites associated with time to event for VISP recurrent stroke or composite vascular event and adjusted for age, sex, and current smoking. Sensitivity models were formed adjusting for treatment and diabetes status. Conditional PH models adjusting for matched pairs was performed on the 22 pairs. Statistical and suggestive significance was determined as p≤5.42e-05 and p≤4.42e-04, respectively (see above for threshold determination) [18].

A one-way analysis of variance (ANOVA) categorizing the genotypes (homozygous-dominant, heterozygous, homozygous-recessive) of 556 variants was implemented on all 305 sphingolipid and fatty acid metabolite levels. A Tukey's HSD post-hoc test was performed to determine differences among genotypes. Statistical and suggestive significance was determined at p≤2.90e-09 (= 0.05/ (556 variants *305 sphingolipid and fatty acid metabolites)) and p≤8.99e-05 (= 0.05/556 variants), respectively.

Weighted Gene Co-expression Network Analysis (WGCNA) [19] was calculated for: 1) clusters of metabolites and clinical traits; 2) clusters of metabolites and DNA methylation as traits; 3) clusters of DNA methylation and metabolite measures as traits; and 4) clusters of DNA methylation and clinical traits. Statistical and suggestive significance was calculated as 1) p≤2.26e-06 (= 0.05/(922 metabolites * 24 traits)) and p≤2.08e-03 (= 0.05/24 traits), respectively; 2) p≤1.14e-10 (= 0.05/(922 metabolites * 473864 loci)) and p ≤1.06e-07 (= 0.05/473864 loci), respectively; 3) p≤1.14e-10 (= 0.05/(922 metabolites * 473864 loci)) and p≤5.42e-05 (= 0.05/922 metabolites), respectively; and 4) p≤4.39e-09 (= 0.05/(473864 loci * 24 traits)) and p≤2.08e-03 (= 0.05/24 traits), respectively.

In the metabolite cluster analyses, modules comprising the 922 metabolites were constructed using a soft-threshold power of 5. Pearson correlations between the first principal component (module eigenvalue) and trait (clinical or methylation beta values) were

performed. WGCNA on the methylation profiles were performed and correlations between these modules and the metabolite profiles or clinical traits used the same quality control steps above. These modules were calculated using the blockwise module function, a soft-threshold power of 14, and a maximum block size of 10,000. Parameters for the module construction for all analyses consisted of the signed topographical overlap matrix type, a signed-hybrid network model, and the minimum number of loci set to 30.

A multiple linear regression analysis adjusting for age, sex, batch effect, current smoking and estimated cellular proportions [20, 21] was performed to identify inferentially methylated CpG loci associated with metabolite measurements. Statistical and suggestive significance was determined at a threshold of p≤1.14e-10 (= 0.05/ (473864 loci*922 metabolites)) and p≤1.06e-07 (= 0.05/473856 CpG loci), respectively. Using the large effect size for regression (f2 = 0.35) with alpha of 0.05, degrees of freedom in the numerator and denominator of 13 and 35, respectively, resulted in a power of 0.6232 [16, 17]. All analyses were performed in R, version 3.5 [22].

## Results

### Baseline demographics for VISP metabolomics subset

Baseline characteristics of VISP study participants with and without recurrent stroke were compared using χ2 and t-tests for categorical and continuous variables, respectively, and are presented in Table 1. Of the 50 participants, 56% (N = 28) suffered a recurrent stroke during the 2-year follow-up. The average baseline age of participants suffering a VISP recurrent stroke was approximately 1.5 years older than those who did not have stroke recurrence at ages 65.07 (standard deviation, SD 11.59) versus 63.68 (9.76), respectively. Those individuals who experienced a recurrent event had worse enrollment stroke severity based on the Modified Rankin Stroke Scale. Only one non-recurrent participant had moderate disability described as a Rankin score of 3, while 11 (39%) of the individuals who experienced stroke recurrence during the trial had moderate disability after the enrollment stroke (p<0.001).

### Identification of metabolites associated with VISP stroke recurrence

Global, untargeted metabolic profiling of 922 metabolites was performed on each of the VISP participant serum samples. A series of statistical tests utilizing the metabolite profiles of VISP participants were performed, including a standard Welch's *t*-test to account for unequal sample sizes and variances within the complete dataset, Cox proportional hazards regression analyses (Cox PH) to reveal associations between the survival time of VISP patients and metabolite detection, and one-sample *t*-tests for matched pair analyses. Collectively, these approaches identified seven significant and 25 suggestive metabolites associated with cardiovascular outcomes or associated clinical traits.

Welch's *t*-test identified 16 metabolites with significant (N = 6) or suggestive (N = 10) associations (Table 2). Of these 16 metabolites, six were associated with recurrent stroke including N-delta-acetylornithine (male only VISP recurrence; p = 1.97e-05), 5α-androstane-3β,17β-diol monosulfate (recurrent stroke ever; p = 2.90e-04), sphingomyelin (d17:1/24:1) (high-dose VISP recurrence; p = 3.00e-04) and ceramide phosphoethanolamine (d18:1/16:0) (high-dose VISP recurrence; p = 3.10e-04). Five significant and five suggestive associations were identified in comparisons of smoking status at trial enrollment. The tobacco and cigarette smoking metabolites cotinine (p = 9.25e-13), 2-naphthol sulfate (p = 2.20e-09), and methylnapththyl sulfate (p = 7.35e-09), were the most significant associations observed.

In matched pair analyses, gamma-glutamylhistidine was identified as a metabolite with observed profile changes between recurrent stroke and non-recurrent participants (fold change, FC = 0.60, p = 3.10e-04) (S5 Table). While not reaching our *a priori* thresholds for

**Table 1. Baseline demographics for VISP metabolomics subset.**

| | VISP | VISP | $P^a$ |
|---|---|---|---|
| | **Nonrecurrent Stroke** | **Recurrent Stroke** | |
| | **(N = 22)** | **(N = 28)** | |
| **Age,** years[b] | 63.68 (9.76) | 65.07 (11.59) | 0.654 |
| **Sex** | | | 0.828 |
| Males | 11 (50.0%) | 16 (57.1%) | |
| Females | 11 (50.0%) | 12 (42.9%) | |
| **Treatment Arm** | | | 0.973 |
| High-dose | 10 (45.5%) | 14 (50.0%) | |
| Low-dose | 12 (54.5%) | 14 (50.0%) | |
| **Current Smoker** | 4 (18.2%) | 7 (25.0%) | 0.815 |
| **Smoker Ever** | 11 (50.0%) | 18 (64.3%) | 0.467 |
| **Cigarettes per day** | 1.50 (3.67) | 2.39 (5.55) | 0.519 |
| **Recurrent stroke ever** | 10 (45.5%) | 28 (100%) | <0.001 |
| **HTN** | 20 (90.9%) | 24 (85.7%) | 0.902 |
| **DM** | 12 (54.5%) | 8 (28.6%) | 0.116 |
| **PNS** | | | 0.600 |
| 0 | 12 (54.5%) | 11 (40.7%) | |
| 1 | 8 (36.4%) | 9 (33.3%) | |
| 2 | 1 (4.5%) | 4 (14.8%) | |
| 3 | 1 (4.5%) | 2 (7.4%) | |
| 4 | 0 (0.0%) | 1 (3.7%) | |
| **Baseline BMI, kg/m²** | 29.70 (6.13) | 28.92 (5.57) | 0.639 |
| **Baseline SBP, mmHg** | 146.41 (22.63) | 143.82 (19.98) | 0.670 |
| **Baseline DBP, mmHg** | 81.64 (9.99) | 79.57 (9.13) | 0.450 |
| **RSS** | | | <0.001 |
| 0 | 0 (0.0%) | 2 (7.1%) | |
| 1 | 13 (59.1%) | 15 (53.6%) | |
| 2 | 8 (36.4%) | 0 (0.00%) | |
| 3 | 1 (4.5%) | 11 (39.3%) | |
| **Days to VISP Stroke** | NA | 256.25 (212.85) | NA |
| **Days to Composite Endpoint** | 552.23 (217.99) | 262.57 (210.46) | <0.001 |

[a]p-value calculated using t-tests and $\chi^2$ for continuous and categorical traits, respectively.

[b]Continuous traits described as mean (SD). Categorical traits described as N (%).

**Abbreviations**: HTN- hypertension; DM- baseline diabetes mellitus status; MI- myocardial infarction; PNS- previous number of strokes; SBP- systolic blood pressure; DBP- diastolic blood pressure; BMI- body mass index; RSS-modified Rankin Stroke Scale.

significance, 18 sphingolipid metabolites had p-values less than 0.05, including sphingomyelin (d18:1/20:1) (FC = 0.77, p = 3.50e-03), sphingomyelin (d18:2/21:0) (FC = 0.66, p = 4.21e-03), and tricosanoyl sphingomyelin (d18:1/23:0) (FC = 0.72, p = 7.16e-03) (S5 Table).

Cox PH survival analyses for the time to recurrent stroke identified tricosanoyl sphingomyelin (d18:1/23:0) (hazard ratio (HR): 0.002 [95% CI:0.000–0.036], p = 1.50e-05), as well as 11 suggestive associations; seven with time to recurrent stroke and four with composite vascular endpoint in the discovery model. Of the suggestively significant metabolites, 10 were associated with sphingomyelin metabolism. Threonate was independently associated with a composite vascular event (HR: 0.001 [0.001–0.135]; p = 4.00e-04) and not related to sphingolipid metabolism. Using global Schoenfeld residual tests, we did not observe any evidence of violation to the proportional hazards assumptions for any of the discovery models (p-values ranging from 0.26

**Table 2. Significant associations from Welch's *t*-test.**

| Phenotype | Metabolite | Pathway | Mean Difference (95% CI) | *t* stat (DF) | p value[a] |
|---|---|---|---|---|---|
| Baseline smoking: yes versus no | cotinine | Tobacco Metabolite | -2.07 (-2.49, -1.65) | -10.01 (43) | 9.25e-13 |
| Baseline smoking: yes versus no | 2-naphthol sulfate | Chemical | -0.77 (-0.97, -0.56) | -7.57 (43) | 2.20e-09 |
| Baseline smoking: yes versus no | methylnaphthyl sulfate | Fatty Acid Metabolism | -0.98 (-1.20, -0.76) | -9.40 (21) | 7.35e-09 |
| Baseline smoking: yes versus no | (2,4 or 2,5)-dimethylphenol sulfate | Food Component/Plant | -0.77 (-1.03, -0.51) | -6.16 (22) | 3.81e-06 |
| Male VISP recurrent stroke: yes versus no | N-delta-acetylornithine | Urea cycle; Arginine and Proline Metabolism | 0.39 (0.24, 0.54) | 5.25 (25) | 1.97e-05 |
| Baseline smoking: yes versus no | o-cresol sulfate | Benzoate Metabolism | -0.81 (-1.13, -0.49) | -5.23 (22) | 3.30e-05 |
| Baseline smoking: yes versus no | 2-ethylphenylsulfate | Benzoate Metabolism | -0.57 (-0.81, -0.34) | -5.28 (15) | 1.10e-04 |
| Recurrent stroke ever: yes versus no | methyl-4-hydroxybenzoate sulfate | Benzoate Metabolism | -0.92 (-1.35, -0.49) | 4.37 (34) | 1.10e-04 |
| Baseline smoking: yes versus no | indoleacetoylcarnitine | Tryptophan Metabolism | -0.54 (-0.78, -0.30) | -4.66 (22) | 1.20e-04 |
| Baseline smoking: yes versus no | hydroxycotinine | Tobacco Metabolite | -1.19 (-1.69, -0.69) | -5.07 (16) | 1.30e-04 |
| No baseline DM VISP recurrence: yes versus no | quinate | Food Component/Plant | -0.97 (-1.45, 0.50) | 4.19 (28) | 2.60e-04 |
| Recurrent stroke ever: yes versus no | 5α-androstan-3β,17β-diol monosulfate | Androgenic Steroids | 0.27 (0.13, 0.41) | -3.93 (45) | 2.90e-04 |
| High-dose trial arm, VISP recurrence: yes versus no | sphingomyelin (d17:1/24:1) | Sphingomyelins | -0.21 (-0.30, -0.11) | 4.45 (19) | 3.00e-04 |
| High-dose trial arm, VISP recurrence: yes versus no | ceramide phosphoethanolamine (d18:1/16:0) | Ceramide PEs | -0.16 (-0.24, -0.08) | 4.44 (19) | 3.10e-04 |
| Baseline smoking: yes versus no | 4-ethylphenylsulfate | Benzoate Metabolism | -0.44 (-0.66, -0.21) | -3.98 (31) | 3.90e-04 |
| Baseline smoking: yes versus no | dimethyl sulfone | Chemical | 0.30 (0.15, 0.45) | 4.14 (23) | 4.20e-04 |

[a] Statistical significance threshold: p≤5.42e-05 (**bold**); suggestive threshold: p≤4.42e-04.

**Abbreviations:** CI- confidence interval; DF- degrees of freedom; PE- phosphoethanolamine.

to 0.537). Sensitivity models including the addition of treatment arm and diabetes status indicated consistent results for these 12 metabolites. A conditional Cox PH model adjusting for pair was performed and resulted in nominal significance for these metabolites (Table 3).

WGCNA analyses of metabolite clusters and clinical traits identified six minimal correlations for 45 VISP participants upon sample filtering (p≤0.01; Fig 1; Table 4). The strongest correlation observed was between a module comprising 77 lipid metabolites (green module) and BMI (r = -0.44, p = 0.002). The yellow module incorporated 81 metabolites, mostly involved in sphingolipid metabolism, and was associated with three stroke outcomes including days to composite vascular endpoint (r = 0.40, p = 0.006), days to VISP recurrent stroke (r = 0.40, p = 0.007), and VISP recurrent stroke status (r = -0.36, p = 0.01).

## Integrative metabolomics-genomics

A one-way ANOVA was performed using the genotypes of 556 variants located within 10 kb upstream/downstream of 24 sphingolipid-metabolism-associated genes selected from the

**Table 3. Significant metabolite associations with time to event.**

| Phenotype | Metabolite | Discovery Model[a] | | | | | Sensitivity Analyses | | |
|---|---|---|---|---|---|---|---|---|---|
| | | B | SE | HR (95% CI) | P[e] | Global Schoenfeld Residual P | Model 2[b] P | Model 3[c] P | Model 4[d] P |
| VISP recurrent stroke | tricosanoyl sphingomyelin (d18:1/23:0) | -6.12 | 1.41 | 0.002 (0.000–0.035) | 1.41E-05 | 0.537 | 1.55e-05 | 4.20E-05 | 5.97E-02 |
| VISP recurrent stroke | sphingomyelin (d18:1/21:0) | -5.17 | 1.29 | 0.005 (0.000–0.071) | 6.21E-05 | 0.390 | 6.75e-05 | 1.06E-04 | 3.46E-02 |
| VISP recurrent stroke | behenoyl sphingomyelin (d18:1/22:0) | -5.40 | 1.36 | 0.005 (0.000–0.064) | 6.92e-05 | 0.472 | 5.74e-05 | 1.70e-04 | 5.63E-02 |
| Composite | tricosanoyl sphingomyelin (d18:1/23:0) | -5.32 | 1.35 | 0.005 (0.000–0.069) | 8.15E-05 | 0.440 | 8.93E-05 | 1.37E-04 | 5.97E-02 |
| Composite | sphingomyelin (d18:1/21:0) | -4.74 | 1.25 | 0.009 (0.001–0.099) | 1.46E-04 | 0.260 | 1.52E-04 | 1.92E-04 | 3.46E-02 |
| VISP recurrent stroke | sphingomyelin (d18:2/23:0) | -5.97 | 1.58 | 0.003 (0.000–0.057) | 1.59E-04 | 0.429 | 1.49E-04 | 3.04E-04 | 2.75E-02 |
| VISP recurrent stroke | sphingomyelin (d17:2/16:0) | -4.87 | 1.30 | 0.008 (0.001–0.098) | 1.73E-04 | 0.350 | 1.73e-04 | 4.75E-04 | 2.24E-02 |
| VISP recurrent stroke | sphingomyelin (d18:2/21:0) | -4.72 | 1.26 | 0.009 (0.001–0.105) | 1.79E-04 | 0.440 | 1.55E-04 | 2.37E-04 | 1.97E-02 |
| VISP recurrent stroke | lignoceroyl sphingomyelin (d18:1/24:0) | -4.51 | 1.22 | 0.011 (0.001–0.121) | 2.25e-04 | 0.365 | 2.03E-04 | 5.72E-04 | 3.81E-02 |
| VISP recurrent stroke | sphingomyelin (d18:1/20:0) | -5.76 | 1.57 | 0.003 (0.000–0.068) | 2.43E-04 | 0.310 | 2.32E-04 | 3.94E-04 | 3.23E-02 |
| Composite | behenoyl sphingomyelin (d18:1/22:0) | -4.66 | 1.29 | 0.009 (0.001–0.122) | 3.15E-04 | 0.383 | 3.13E-04 | 4.85E-04 | 5.63E-02 |
| Composite | threonate | -4.49 | 1.26 | 0.011 (0.001–0.133) | 3.70E-04 | 0.390 | 3.77E-04 | 5.84E-04 | 3.86E-02 |

[a] Discovery model adjusts for age, sex, current smoking.

[b] Model 2 adjusts for discovery model plus treatment arm.

[c] Model 3 adjusts for model 2 plus diabetes mellitus status.

[d] Model 4 is a conditional Cox proportional hazards model that adjusts for pair.

[e] Statistical significance threshold: p≤5.42e-05; suggestive threshold: p≤4.42e-04.

**Abbreviations:** B-beta coefficient; SE- standard error; HR- hazard ratio; CI- confidence interval; p- p-value.

Harmonizome database [23] and the profiles of 305 sphingolipid/fatty acid metabolites (S3 Table). These analyses identified 29 variant-metabolite associations between 16 unique metabolites and 23 variants, that exceeded a suggestive threshold of p≤8.99e-05. Furthermore, Tukey's post-hoc tests identified 54 differences among genotypes (Table 5). The most significant association identified was between leukotriene B4 and rs7025659 (p = 6.12e-07; post-hoc comparison between TG-GG genotypes p = 3.00E-07), an upstream variant of *SPTLC1* (serine palmitoyltransferase long chain base subunit 1). Sphingosine and spinganine were nominally associated with five common variants and one distinct variant (rs10757056 (sphingosine only), rs10118089, rs7020745, rs10125228, rs10757058, rs10118371) in the region of *ACER2* encoding alkaline ceramidase 2 (Table 5).

## Integrative metabolomics-epigenomics

WGCNA analyses of methylation modules and 39 clinical traits identified 14 associations that met or exceed p≤6.25e-06, of which nine were statistically significant (p≤2.71e-09). The two most significant associations were with number of strokes prior to enrollment and the thistle4 module (r = 0.91; p = 2.00e-18) and lightsteelblue1 module (r = -0.89; p = 6.00e-17). Other

A.

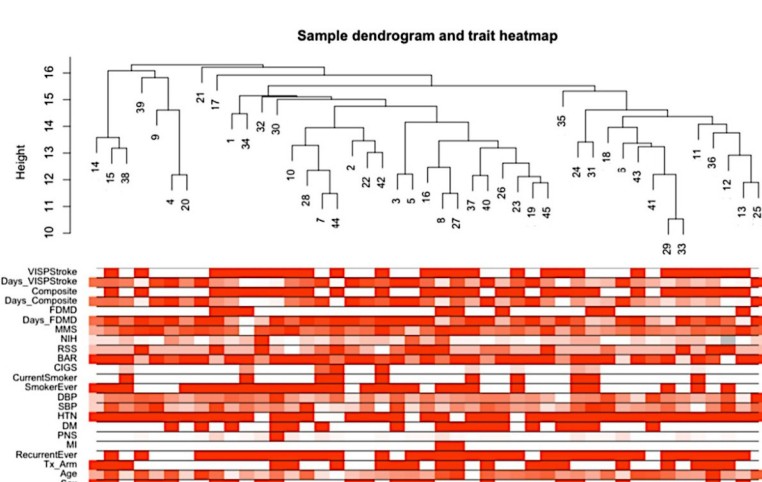

B.

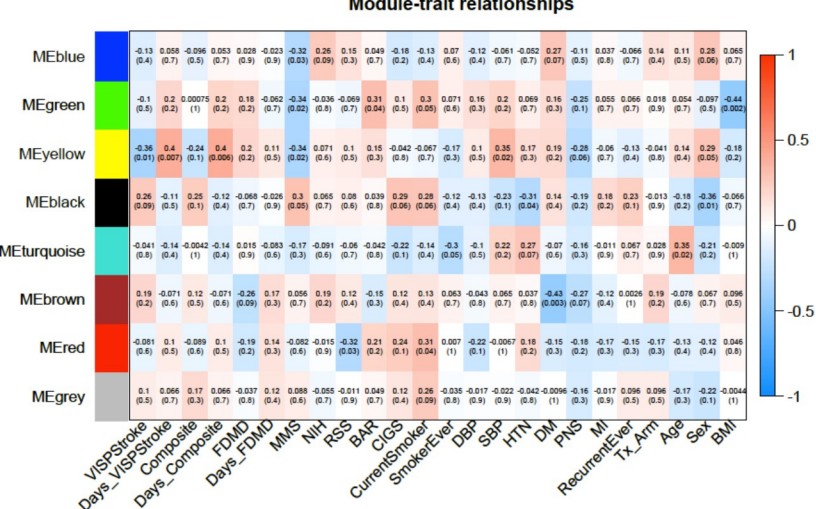

**Fig 1. WGCNA for log transformed metabolite profiles of 45 AFR.** A.) Sample dendrogram and trait heatmap of metabolites and clinical traits/stroke outcomes B.) Module-trait heatmap of module eigenvalues and stroke outcomes.

associations were detected with clinical biomarkers including baseline plasma folate, pro-thrombin fragments F1 + 2, thrombin-antithrombin complex, thrombomodulin, and triglycerides (S6 Table).

WGCNA of methylation modules and metabolites identified 38 associations, of which 24 (22 significant and 2 suggestive) were between modules of methylation loci and pharmacological sub-pathways including analgesics/anesthetics (n = 7), cardiovascular (n = 6), gastrointestinal (n = 1), neurological (n = 2), psychoactive (n = 4), respiratory (n = 2), and topical agents (n = 2). Additional significant associations were found between the eicosanoid 5-hydroxyeicosatetraenoic acid and the brown4 (177 loci, r = -0.90, p = 2.00e-22) and lightpink3 modules (77 loci, r = 0.92, p = 3.00e-20), as well as threonylphenylalanine and the darkolivegreen (284 loci, r = -0.90, p = 5.00e-21) and the lightsteelblue1 (214 loci, r = 0.89, p = 8.00e-17) modules (S7 Table).

**Table 4. Top WGCNA module-trait associations for metabolite profiles and stroke traits/outcomes.**

| Module | # of Metabolites/Module | Pathway[a] | Correlated Trait | r | P[a] |
|---|---|---|---|---|---|
| green | 77 | Lipid metabolism (plasmalogen) | BMI | -0.44 | 0.002 |
| brown | 103 | Amino acid metabolism (BCAA); Lipid metabolism (PC; PE) | DM | -0.43 | 0.003 |
| yellow | 81 | Lipid metabolism (sphingomyelins) | Days to Composite Endpoint | 0.40 | 0.006 |
| yellow | 81 | Lipid metabolism (sphingomyelins) | Days to VISP Stroke | 0.40 | 0.007 |
| black | 38 | Lipid metabolism (androgenic, progestin steroids) | Sex | -0.36 | 0.01 |
| yellow | 81 | Lipid metabolism (sphingomyelins) | VISP Stroke | -0.36 | 0.01 |

[a] Statistical significance threshold p≤2.26e-06; suggestive threshold: p≤2.08e-03.

**Abbreviations:** BCAA- branched chain amino acids; PC- phosphatidylcholine; PE- phosphatidylethanolamine; BMI- body mass index, kg/m$^2$; DM-diabetes mellitus status

Network associations between metabolite modules and methylation loci as traits detected 64 suggestive associations (p≤5.42e-05), with the strongest correlation observed between the green module (lysophospholipids and phosphatidylcholine synthesis) and cg18461635, a methylation locus in an intron of *CCDC12* (r = -0.65, p = 2.00e-06) (S8 Table).

For tests of epigenetic-metabolite associations, linear regression analyses identified 2,622 methylation loci that met or exceeded p≤1.05e-07 (2-naphthol sulfate (n = 245), (2,4 or 2,5) dimethylphenol sulfate (n = 2,023), o-cresol-sulfate (n = 192), 2-ethylphenyl sulfate (n = 130), cotinine (n = 31), and hydroxycotinine (n = 1))(S9 Table).

## Discussion

This study integrated metabolomics, epigenomics, and genomics data in analyses of recurrent stroke in AAs. The metabolites identified in the baseline smoking *t*-test analysis included those present in tobacco or cigarette smoke. Cotinine, an alkaloid found in tobacco leaves and the main metabolite of nicotine, [24] was the most significant metabolite in a series of group-means comparisons. Serum cotinine levels have been used as an indicator of second-hand smoke exposure where high second-hand smoke levels (serum cotinine >0.7 ng/mL) increased coronary heart disease risk up to 1.5-fold [25, 26]. In total, ten metabolites were detected in our smoking analyses. While these associations were significant, they were not surprising and serve as a proof-of-concept, validating the utility of metabolomics in the VISP study. Notable differences in the levels of N-delta-acetylornithine, sphingomyelin (d17:1/24:1), and ceramide phosphoethanolamine (d18:1/16:0) were observed in *t*-tests of stroke recurrence. N-delta-acetylornithine was previously associated with *NAT8*, a gene correlated with creatinine levels and chronic kidney disease in AA [27]. This is of interest since the risk of stroke is 5–30 times higher in patients with chronic kidney disease [28]. Identified in the matched pairs analysis, gamma-glutamylhistidine is a component of gamma-glutamyl amino acid metabolism. Previous studies suggested elevated gamma-glutamyl transferase is associated with increased risk of stroke, coronary heart disease, arterial HTN, and cardiovascular disease-related mortality [29].

Sphingolipid-related metabolites were implicated in Welch's *t*-tests, survival analyses, matched-pairs analysis, and WGCNA, where we observed an increase in sphingomyelin levels to confer a level of neuroprotection or delayed event recurrence. Sphingolipids play a vital role in intracellular signal transduction and regulation of cellular proliferation, maturation, apoptosis, and cellular stress response, as well as being components of the cardiomyocyte cell membrane [30]. Inflammatory cytokines, such as TNF-α, may induce the synthesis of ceramides from sphingomyelins via sphingomyelinase [31]. While numerous studies report detrimental effects associated with increased ceramide levels, sphingosine-1-phosphate (S1P), has a

**Table 5. Comparison between sphingolipid-associated variant genotype calls and mean metabolite level.** Sorted by ANOVA P value.

| Metabolite | Variant | Closest Gene | ANOVA P[a] | Genotype Comparison | | Tukey HSD P |
|---|---|---|---|---|---|---|
| leukotriene B4 | rs7025659 | SPTLC1 | 6.12E-07 | TG | GG | 3.00E-07 |
| | | | | TT | TG | 7.24E-04 |
| 5-HETE | rs11601088 | SMPD1 | 2.15E-05 | GG | AG | 2.15E-05 |
| 1-linoleoyl-GPE (18:2) | rs2898458 | ASAH1 | 2.24E-05 | GA | AA | 1.69E-04 |
| | | | | GG | GA | 4.72E-04 |
| octadecadienedioate (C18:2-DC) | rs918957 | CERS6-AS1 | 2.59E-05 | CC | CA | 1.79E-05 |
| | | | | CC | AA | 1.24E-04 |
| 3-hydroxyoctanoate | rs7897345 | ASAH2B | 2.80E-05 | CT | CC | 1.91E-05 |
| | | | | TT | CC | 8.11E-05 |
| 3-hydroxyoctanoate | rs13376734 | ASAH2B | 2.80E-05 | TT | CC | 8.11E-05 |
| | | | | TT | TC | 1.91E-05 |
| 2-hydroxyglutarate | rs1124626 | CERS4 | 3.12E-05 | AG | AA | 5.13E-05 |
| | | | | GG | AA | 1.75E-05 |
| 2-hydroxyglutarate | rs11666971 | CERS4 | 3.31E-05 | GG | AA | 1.87E-05 |
| | | | | GG | GA | 5.81E-05 |
| sphingosine | rs10757056 | ACER2 | 3.41E-05 | TT | CC | 4.14E-04 |
| | | | | TT | TC | 3.76E-05 |
| sphingosine | rs10118089 | ACER2 | 3.48E-05 | AG | AA | 3.04E-05 |
| | | | | GG | AA | 3.67E-04 |
| sphingosine | rs7020745 | ACER2 | 3.48E-05 | AC | AA | 3.04E-05 |
| | | | | CC | AA | 3.67E-04 |
| 1-palmitoyl-GPG (16:0) | rs135700 | CERK | 3.57E-05 | TC | CC | 9.63E-05 |
| | | | | TT | CC | 3.76E-02 |
| 1-oleoyl-GPE (18:1) | rs2898458 | ASAH1 | 3.63E-05 | GA | AA | 2.71E-04 |
| | | | | GG | GA | 6.25E-04 |
| sphinganine | rs10118089 | ACER2 | 5.07E-05 | AG | AA | 3.32E-05 |
| | | | | GG | AA | 2.25E-04 |
| sphinganine | rs7020745 | ACER2 | 5.07E-05 | AC | AA | 3.32E-05 |
| | | | | CC | AA | 2.25E-04 |
| (S)-3-hydroxybutyrylcarnitine | rs12358176 | SGMS1 | 5.11E-05 | AG | AA | 1.33E-04 |
| | | | | GG | AA | 3.74E-05 |
| malonate | rs12355439 | SGMS1 | 5.57E-05 | CT | CC | 5.47E-05 |
| | | | | TT | CC | 3.27E-05 |
| malonate | rs11595661 | SGMS1 | 5.57E-05 | GG | AA | 3.27E-05 |
| | | | | GG | GA | 5.47E-05 |
| palmitoyl-oleoyl-glycerol (16:0/18:1) | rs1466447 | CERS4 | 5.72E-05 | TT | CT | 3.40E-05 |
| sphingosine | rs10125228 | ACER2 | 6.06E-05 | AG | AA | 5.04E-05 |
| | | | | GG | AA | 4.46E-04 |
| sphinganine | rs10125228 | ACER2 | 6.41E-05 | AG | AA | 4.39E-05 |
| | | | | GG | AA | 2.78E-04 |
| sphingosine | rs10757058 | ACER2 | 6.96E-05 | CT | CC | 4.90E-05 |
| | | | | TT | CC | 4.24E-04 |
| sphingosine | rs10118371 | ACER2 | 6.96E-05 | TT | CC | 4.24E-04 |
| | | | | TT | TC | 4.90E-05 |
| sphinganine | rs10757058 | ACER2 | 7.62E-05 | CT | CC | 4.66E-05 |
| | | | | TT | CC | 2.68E-04 |

(*Continued*)

**Table 5.** (Continued)

| Metabolite | Variant | Closest Gene | ANOVA P[a] | Genotype Comparison | | Tukey HSD P |
|---|---|---|---|---|---|---|
| sphinganine | rs10118371 | *ACER2* | 7.62E-05 | TT | CC | 2.68E-04 |
| | | | | TT | TC | 4.66E-05 |
| 1-oleoyl-2-arachidonoyl-GP (18:1/20:4) | rs773280 | *SPTLC1* | 7.67E-05 | TG | GG | 7.67E-05 |
| laurate (12:0) | rs10763500 | *SGMS1* | 7.68E-05 | TT | CC | 6.01E-05 |
| | | | | CT | CC | 2.39E-04 |
| leukotriene B4 | rs6721420 | *CERS6* | 7.81E-05 | TT | CT | 7.81E-05 |
| ceramide phosphoethanolamine (d18:1/16:0) | rs1047123 | *CERK* | 8.63E-05 | GC | CC | 8.09E-05 |
| | | | | GG | GC | 3.31E-02 |

[a] Statistical significance threshold: p≤2.90e-07; suggestive significance threshold: p≤8.99e-05.

**Abbreviations**: HSD- Honestly Significant Difference; *ACER2*- Alkaline Ceramidase 2; *ASAH1*- N-Acylsphingosine Amidohydrolase 1; *ASAH2B*- N-Acylsphingosine Amidohydrolase 2B; *CERK*- Ceramide Kinase; *CERS4*- Ceramide Synthase 4; *CERS6*- Ceramide Synthase 6; *CERS6-AS1*- CERS6 Antisense RNA 1; *SGMS1*- Sphingomyelin Synthase 1; *SMPD1*- Sphingomyelin Phosphodiesterase 1; *SPTLC1*- Serine Palmitoyltransferase Long Chain Base Subunit 1.

neuroprotective function during ischemia [32]. This is presumably due to S1P regulating pro-survival mechanisms through the suppression of pro-apoptotic factors including caspase 3 and the activation of protein kinase B or Akt [32]. Furthermore, increased ceramide levels have been reported in patients with HTN, with ceramide concentrations positively correlated with HTN severity [33]. Further investigation on the homeostatic balance between sphingomyelins and ceramides are needed to validate biological markers of stroke recurrence and stroke-related comorbidities such as HTN.

Examining variants within sphingolipid metabolism enzymes identified 23 unique variants that were associated with metabolites within sphingolipid and fatty acid metabolism. In the present study, rs7025659 was associated with leukotriene B4, a pro-inflammatory lipid mediator derived from arachidonic acid [34]. Leukotriene B4 levels are associated with poorer functional recovery in ischemic stroke patients. A 2020 study reported that higher leukotriene B4 levels on days 0 and 7 post-ischemic stroke are associated with poorer functional recovery based on RSS [34]. Five genetic variants within *ACER2* were associated with plasma sphinganine and sphingosine levels, while an additional *ACER2* variant (rs10757056) was associated with sphingosine levels. *ACER2* encodes alkaline ceramidase 2, which hydrolyzes ceramides to generate sphingosine and sphingosine-1-phosphate [35]. *ACER2* expression is in-part regulated by the hypoxia-inducible factor 2α, an atherosclerosis suppressor and known ischemic stroke marker [36, 37]. Additionally, four variants within *SGMS1*, rs10763500, rs11595661, rs12355439, and rs12358176, were implicated in the ANOVA. *SGMS1* encodes the sphingomyelin synthase 1 protein, which is a transmembrane protein highly expressed in the brain [38] and functions by metabolizing ceramide into sphingomyelin [39]. While these results are promising, further studies with increased sample sizes are needed to determine the biological implications of these variants in regard to metabolism and stroke recurrence.

Integrative analyses consisting of methylation, metabolite profiles, and covariates have reinforced our univariate results implicating tobacco and smoking related metabolites. This finding could strengthen the paradigm that secondhand smoke influences stroke and stroke recurrence [40]. Network analyses identified clusters of DNA methylation loci that were correlated with concentrations of prescribed analgesics, cardiovascular, neurological, and psychoactive drugs. These associations are to be expected in a population with individuals having suffered prior strokes and cardiovascular events, as these drugs are commonly prescribed to post-stroke patients. It is possible these drug associations could be useful for identifying

individuals who are poor drug metabolizers or indicative of the efficacy and/or potency of particular therapeutics.

This study was performed in a subset of AA VISP clinical trial participants, a population more likely to experience a recurrent stroke than their white counterparts. Furthermore, risk factors for stroke, specifically HTN, diabetes mellitus, and chronic kidney disease, are more prevalent in this population. The recurrent stroke phenotype overall is understudied and its etiology is not well understood; therefore, the multi-omic analyses on this complex disease is a strength of this study. DNA used in the methylation analyses was extracted from whole blood samples upon enrollment in the trial. Although not optimal due to cellular heterogeneity, whole blood provides a valuable resource that is available for replication studies. Adjusting for this limitation, cellular proportions were calculated *in silico*, and used as covariates in the association models.

This study has a modest sample size of 50 individuals and thus our statistical power was limited and analyses could only confidently detect large effect associations. This was a primary drawback in addition to the lack of validation. Studies of AA with global metabolite, methylation, and genetic data, as well as adjudicated recurrent stroke outcomes are rare, further limiting our ability to include a replication cohort. Although stroke subtype is not adjudicated in VISP, these stroke cases most likely represent small vessel variety due to the inclusion/exclusion criteria and higher proportion of small vessel (lacunar) ischemic stroke typically overserved in AA [41].

While the design of matched pairs is an optimal approach, in our analyses larger sample sizes could improve the detection of associations of small and/or medium effect size. Multiple blood samples per individual would allow for more comprehensive matched-pairs analyses, while matching based on similar pharmacological profiles and time of blood draw would be ideal as the consumption of pharmaceutical drugs, time of day, and time of year (i.e. season), all influence metabolite levels. We also cannot conclusively state that the metabolite levels were not altered by time post-stroke since the metabolites were collected within 120 days of stroke onset. Potentially acute metabolite profiles could differ from sub-acute or chronic stroke metabolite profiles. Additionally, samples should be matched closer to age, as our paired individuals were within eight years of age.

In conclusion, even with a limited sample size, findings from this study provide insight into associations between metabolites, DNA methylation, genetic variants and recurrent stroke, thus identifying potential plasma biomarkers in AA. Further studies are needed to highlight the pathways underlying these associations in regard to their biological and clinical applications.

## Supporting information

**S1 Appendix.**
(XLSX)

**S1 Table. Baseline demographics of matched pairs.**
(DOCX)

**S2 Table. Demographics by pair.**
(DOCX)

**S3 Table. Sphingolipid related genes used in SNP-based tests.** Gene region based on hg19.
(DOCX)

**S4 Table. Summary of Welch's *t*-test comparisons.**
(DOCX)

**S5 Table. One-sample *t*-test of the fold change from 22 matched pairs.**
(DOCX)

**S6 Table. WGCNA module-trait associations for methylation profiles and stroke traits/ outcomes.**
(DOCX)

**S7 Table. Significant WGCNA module (DNA methylation)-trait (metabolite) associations.**
(DOCX)

**S8 Table. WGCNA module (metabolite)-trait (DNA methylation loci) associations.**
(DOCX)

**S9 Table. Summary of significant metabolites-methylation associations using fully adjusted regression models.**
(DOCX)

# Acknowledgments

The authors would like to thank the individuals who participated in the VISP studies. We would additionally like to thank Gregory R. Wagner, PhD and Metabolon for generation of the metabolite data. Michele M. Sale passed away before the submission of the final version of this manuscript. The corresponding author (KLK) accepts responsibility for the integrity and validity of the data collected and analyzed.

# Author Contributions

**Conceptualization:** Michèle M. Sale, Bradford B. Worrall, Keith L. Keene.

**Data curation:** Nicole M. Davis Armstrong, Wei-Min Chen, Fang-Chi Hsu, Michael S. Brewer, Keith L. Keene.

**Formal analysis:** Nicole M. Davis Armstrong, Kelsey J. Spragley, Wei-Min Chen, Fang-Chi Hsu, Michael S. Brewer.

**Funding acquisition:** Michèle M. Sale, Bradford B. Worrall, Keith L. Keene.

**Methodology:** Nicole M. Davis Armstrong, Wei-Min Chen, Fang-Chi Hsu, Michael S. Brewer, Patrick J. Horn, Stephen R. Williams, Michèle M. Sale, Bradford B. Worrall, Keith L. Keene.

**Supervision:** Keith L. Keene.

**Visualization:** Nicole M. Davis Armstrong, Kelsey J. Spragley, Wei-Min Chen, Fang-Chi Hsu, Michael S. Brewer, Patrick J. Horn, Stephen R. Williams.

**Writing – original draft:** Nicole M. Davis Armstrong, Kelsey J. Spragley.

**Writing – review & editing:** Wei-Min Chen, Fang-Chi Hsu, Michael S. Brewer, Patrick J. Horn, Stephen R. Williams, Bradford B. Worrall, Keith L. Keene.

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
