## [Decision Letter · Decision Letter 0]

24 Dec 2020

PONE-D-20-35612

Multi-omic analysis of stroke recurrence in African Americans from the Vitamin Intervention for Stroke Prevention (VISP) clinical trial

PLOS ONE

Dear Dr. Keene,

Thank you for submitting your manuscript to PLOS ONE. After careful consideration, we feel that it has merit but does not fully meet PLOS ONE’s publication criteria as it currently stands. Therefore, we invite you to submit a revised version of the manuscript that addresses the points raised during the review process.

We look forward to receiving your revised manuscript.

Kind regards,

Yiqiang Zhan

Academic Editor

PLOS ONE

Reviewers' comments:

Reviewer's Responses to Questions

**Comments to the Author**

1. Is the manuscript technically sound, and do the data support the conclusions?

Reviewer #1: Yes

Reviewer #2: Yes

2. Has the statistical analysis been performed appropriately and rigorously? 

Reviewer #1: Yes

Reviewer #2: Yes

3. Have the authors made all data underlying the findings in their manuscript fully available?

Reviewer #1: Yes

Reviewer #2: Yes

4. Is the manuscript presented in an intelligible fashion and written in standard English?

Reviewer #1: Yes

Reviewer #2: Yes

5. Review Comments to the Author

Reviewer #1: This study suggests results that are helpful in finding biomarkers related to stroke recurrence in high-risk groups.

To further improve this research method, it would be better to describe the type of ischemic stroke or the lesion of the stroke in the subjects of this study. The relationship between these and biomarkers could lead to better research results.

If this is not possible, it would be better to present it as a further study in the discussion.

Reviewer #2: This manuscript is showing characteristic candidates of stroke based on metabolomics, genetic variations and epigenomics. statisitc analysis was carefully performed and the results showed interesting insights. Therefore, it is very excellent and acceptable for this journal.

6. PLOS authors have the option to publish the peer review history of their article (what does this mean?). If published, this will include your full peer review and any attached files.

Reviewer #1: **Yes: **Seung-Bo Yang

Reviewer #2: **Yes: **Kazuo Ishii

---

## [Author Response · Author response to Decision Letter 0]

2 Feb 2021

Reviewer #1: This study suggests results that are helpful in finding biomarkers related to stroke recurrence in high-risk groups.

To further improve this research method, it would be better to describe the type of ischemic stroke or the lesion of the stroke in the subjects of this study. The relationship between these and biomarkers could lead to better research results.

If this is not possible, it would be better to present it as a further study in the discussion.

Reviewer #2: This manuscript is showing characteristic candidates of stroke based on metabolomics, genetic variations and epigenomics. statisitc analysis was carefully performed and the results showed interesting insights. Therefore, it is very excellent and acceptable for this journal.

Response: Based on the inclusion and exclusion criteria for the VISP trial, the current study focuses on nondisabling ischemic stroke. The study specifically excluded those with stroke due to cardioembolic source including atrial fibrillation, arterial dissection, and moderate to severe carotid stenosis considered for surgical revascularization. Although ischemic stroke subtyping is not available for participants in VISP, the study population is comprised of patients with stroke primarily due to small vessel occlusion or intracranial large artery atherosclerosis with some individuals with non-surgical extracranial large artery atherosclerosis and cryptogenic/ undetermined stroke subtypes. We have modified the methods section by adding the inclusion/exclusion criteria and modified the discussion to reflect our statement above.

---

## [Editor Report · Decision Letter 1]

4 Feb 2021

Multi-omic analysis of stroke recurrence in African Americans from the Vitamin Intervention for Stroke Prevention (VISP) clinical trial

PONE-D-20-35612R1

Dear Dr. Keene,

We’re pleased to inform you that your manuscript has been judged scientifically suitable for publication and will be formally accepted for publication once it meets all outstanding technical requirements.

Kind regards,

Yiqiang Zhan

Academic Editor

PLOS ONE
---

## [Editor Report · Acceptance letter]

22 Feb 2021

PONE-D-20-35612R1 

Multi-omic analysis of stroke recurrence in African Americans from the Vitamin Intervention for Stroke Prevention (VISP) clinical trial 

Dear Dr. Keene:

I'm pleased to inform you that your manuscript has been deemed suitable for publication in PLOS ONE. Congratulations! Your manuscript is now with our production department. 

Kind regards, 

on behalf of

Dr. Frank Y Zhan 

Academic Editor

PLOS ONE